# Young Adults with Higher Salt Intake Have Inferior Hydration Status: A Cross-Sectional Study

**DOI:** 10.3390/nu14020287

**Published:** 2022-01-11

**Authors:** Jianfen Zhang, Na Zhang, Shufang Liu, Songming Du, Guansheng Ma

**Affiliations:** 1Department of Nutrition and Food Hygiene, School of Public Health, Peking University, 38 Xue Yuan Road, Haidian District, Beijing 100191, China; ZJF@bjmu.edu.cn (J.Z.); mags@bjmu.edu.cn (G.M.); 2Laboratory of Toxicological Research and Risk Assessment for Food Safety, Peking University, 38 Xue Yuan Road, Haidian District, Beijing 100191, China; 3School of Public Health, Hebei University Health Science Center, 342 Yuhua Road, Lianchi District, Baoding 071000, China; shufangliu@126.com; 4Chinese Nutrition Society, Room 1405, Beijing Broadcasting Building, No. 14 Jianguomenwai Street, Chaoyang District, Beijing 100053, China; dusm9709@126.com

**Keywords:** hydration biomarkers, water intake patterns, drinking patterns, salt intake

## Abstract

The body’s water and sodium balances are tightly regulated and monitored by the brain. Few studies have explored the relationship between water and salt intake, and whether sodium intake with different levels of fluid intake leads to changes in hydration status remains unknown. The aim of the present study was to determine the patterns of water intake and hydration status among young adults with different levels of daily salt intakes. Participants’ total drinking fluids and water from food were determined by a 7-day 24-h fluid intake questionnaire for 7 days (from Day 1 to Day 7) and duplicate portion method (Day 5, Day 6 and Day 7). Urine of 24 h for 3 days (Day 5, Day 6 and Day 7) was collected and tested for the osmolality, the urine-specific gravity (USG), the concentrations of electrolytes, pH, creatinine, uric acid and the urea. The fasting blood samples for 1 day (Day 6) were collected and measured for the osmolality and the concentrations of electrolytes. The salt intakes of the participants were evaluated from the concentrations of Na of 24 h urine of 3 days (Day 5, Day 6 and Day 7). Participants were divided into four groups according to the quartile of salt intake, including the low salt intake (LS_1_), LS_2_, high salt intake (HS_1_) and HS_2_ groups. In total, 156 participants (including 80 male and 76 female young adults) completed the study. The salt intakes were 7.6, 10.9, 14.7 and 22.4 g among participants in the four groups (LS_1_, LS_2_, HS_1_ and HS_2_ groups, respectively), which differed significantly in all groups (*F* = 252.020; all *p* < 0.05). Compared to the LS_1_ and LS_2_ groups, the HS_2_ group had 310–381, 250–358 and 382–655 mL more amounts of water from the total water intake (TWI), total drinking fluids and water from food (all *p* < 0.05), respectively. Participants in the HS_2_ group had 384–403, 129–228 and 81–114 mL more in the water, water from dishes and staple foods, respectively, than those in the groups of LS_1_ and LS_2_ (*p* < 0.05). The HS_2_ group excreted 386–793 mL more urine than those in the groups of LS_1_ and LS_2_ (*p* < 0.05). However, regarding urine osmolality, the percentage of participants with optimal hydration status decreased from 41.0% in LS_1_ and LS_2_ to 25.6% in the HS_2_ group (*p <* 0.05). Participants with higher salt intake had higher TWI, total drinking fluids and water from food. Nevertheless, they had inferior hydration status. A reduction in salt intake should be encouraged among young adults to maintain optimal hydration status.

## 1. Introduction

The body’s water and sodium balances are tightly regulated and monitored by the brain, including thirst, the regulation of water, electrolyte excretion or retention by the kidneys [1]. Physiological regulation of the intake and output of sodium and water is necessary to maintain homeostasis. The dynamic balance of the water defines hydration as either dehydration or overhydration. Studies have shown that dehydration, which was defined as less water intake than water loss, impeded cognitive performances, attenuating vigilance, attention and working memory [2]. Furthermore, overhydration, defined as more water intake than water loss, can result in headache, nausea and memory loss [3]. Therefore, it is crucial to have adequate water intake to maintain optimal hydration status. Moreover, it is known that salt is a combination of sodium and chloride. To maintain vital functions, including proper blood volume and blood pressure, human cells require about 0.5 g/day of sodium. Moreover, the main physiological function of sodium is to maintain positively charged ions in extracellular fluid, which participate in water metabolism, ensure the balance of water in the body and maintain the balance of acid and alkali in the body. Studies have shown that insufficient water intake increased the plasma salt level and osmolality. High intake of salt and low intake of water increased the extracellular osmolality which would lead to hypertension, particularly in salt-sensitive individuals. Furthermore, high sodium or salt intake has also been linked with a high risk of cardiovascular diseases in the general population [4,5]. Studies have demonstrated that water and sodium deficits are predisposing factors for exertional heat stroke and exertional heat exhaustion [6]. Therefore, it is of vital importance to maintain the proper consumption of water and sodium for the human body.

The recommendation of the sodium intake in the World Health Organization (WHO) for the prevention of chronic disease is below 2000 mg/d [7]. Yet only a few countries have an average daily sodium intake that does not exceed the recommended level [8]. Regarding the sources of sodium, most food preservatives have high sodium content and are major causes of increased dietary intake of sodium. Consequently, a study showed that the main contributor to daily sodium intake among Chinese adults was the salt added to food [9]. Among Asian Americans, half of the sodium consumed came from ten food categories, including processed meats [10]. Similarly, in Australia, the main sources are breads, processed meats and savory sauces [11]. Therefore, types of food may influence sodium intake. Among studies investigating the association between water intake and sodium intake, the conclusions were controversial. Generally, the consensus is that eating salty food stimulates thirst. After ingestion of the sodium from foods or fluids, there would be a rise in the concentration of sodium in plasma; therefore, in order to maintain fluid homeostasis, thirst was stimulated to promote individuals to drink. However, a study concluded that increased salt consumption resulted in body water conservation and the decrease in fluid intake among 10 healthy men [12]. Another study showed that varying salt intake from 0.6 g/d to 24 g/d may not change the fluid intake or urine volume in humans [13,14]. Whether the differences in sodium intake could influence the intake of total drinking fluids or water from food remains to be further studied. Along with studies conducted among humans, the results of some experimental studies in animals were not consistent, either. A randomized controlled trial conducted among cats showed that the increase of dietary sodium intake led to the increase in water intake and the volume of urine [15]. However, in minks, the addition of 0.5% NaCl intake did not promote the intake of water [16]. Unfortunately, the types of fluid intake and hydration biomarkers (including urine osmolality, urine specific gravity and plasma osmolality) were not explored in the above researches.

Moreover, studies reporting the association between sodium intake and hydration status are scarce, especially among people in free-living conditions. A randomized crossover study exploring the relationship between hydration status, thirst and the preference for salt among adults showed that hydration status led to changes in salt desire and food with different water contents [17]. Notwithstanding, for children, salt intake did not affect hydration status [18]. Therefore, whether the sodium intake could lead to changes in fluid intake and affect the hydration status remains unknown. In China, the sodium intake is 5013 mg/day, which is greater than the WHO recommendation and the physiological requirements of the human body. Nevertheless, in China, the differences in hydration biomarkers among participants with different salt intake have not yet been explored. Water intake patterns, including the drinking patterns (including the amounts and types of fluids in the present study) and food intake pattern (including the amounts of water from different types of food consumed in the present study), have been related to health. Studies demonstrated that sodium intake may affect types of fluid intake. Data from children in UK and Australia revealed a positive association between sodium intake and sugar-sweetened beverages (SSBs) intake [19,20]. An increasing number of studies asserted that positive associations between SSBs and the risk of weight gain, metabolic syndrome or obesity-related cancers were found in adults, children and adolescents [21,22,23,24]. Additionally, the intake of milk was associated with lower all-cause mortality in men [25]. Hence, the types of total drinking fluids are required for exploration. In reference to the volumes of water from food, the contributions of water from food to TWI are different among countries [26,27,28]. In reference to the young adults with different habitual total drinking fluids intake in China, nearly 40.9–61.3% of the TWI came from food [29]; however, the drinking patterns and food intake patterns among young adults with different levels of salt intake have not been reported. Taken together, these studies suggest that the differences in the water intake patterns and hydration biomarkers between participants with different salt intake are unclear. Therefore, careful assessment of the responses of the water intake patterns and hydration biomarkers to high dietary salt in healthy young adults, both males and females, is warranted.

The aims of the current study were to investigate the differences in the water intake patterns, including the drinking patterns and the food intake patterns, in young adults with different levels of salt intake in free-living conditions. Second, we explored the differences in hydration biomarkers among them. This will lead to health education for young adults about fluid intake and sodium intake.

## 2. Methods

### 2.1. Study Design

A cross-sectional study over 7 consecutive days was implemented. The period of the recruitment of the participants was from 15 March 2017 to 30 March 2017, at one college.

### 2.2. Sample Size Calculation

In one study conducted among adults [30], the detection rate (*p*) of the adults with the intake of salt exceeding the recommendation of Chinese dietary guidelines of 6 g/d was 0.7; then, setting the *d* = 0.11, α = 0.05, Z_α_ = 1.96, the sample size was calculated using the formula as followed:n= Z^2^_α_*p* (1 − *p*)/*d*^2^

In total, 136 participants were needed. In the present study, 159 participants were recruited.

### 2.3. Participants

The inclusion criteria were that participants including male and female adults were healthy and aged 18–23 years [31]. For participants who were excluded from the study, detailed description was given in the previous study [26]. The flow chart of participants is shown in Figure 1.

### 2.4. Study Procedure

On Day 1, all the subjects were instructed to have anthropometric measurements taken, with the height and weight included, by trained investigators with standard methods. Then, participants were supervised to fulfill the self-designed 7-day 24-h fluid intake questionnaire to record their fluid intake every day over 7 days with the guidance of the investigators. Furthermore, all the food the participants ate during the three consecutive days (including the two weekdays and one weekend day, Day 5, Day 6 and Day 7) over the 7 days was weighed and recorded. Moreover, during the three consecutive days, the 24 h urine samples, including the first morning urine, were collected. On the morning of Day 6, the fasting blood sample collection was performed. All the participants were asked to live as usual, and the assessments for participants were performed in the lab of the university by trained investigators. The study procedure is shown in Figure 2.

### 2.5. Measurement of Total Water Intake

The total drinking fluids were assessed by the self-designed 7-day 24-h fluid intake record questionnaire, which has been used in our previous studies [32,33]. Furthermore, every participant was asked to drink any fluids using the cups that were supplied to them, which were accurate to the nearest 5 mL. Each time they drank, they were also required to record the type, the amount and the place of drinking fluid, which has been described in our previous study [29]. About the General Standard for Beverages of China (GB/T 1-789-2015) [34], all the drinking fluids were classified into water, tea, milk and milk products, SSBs, alcohols and other beverages [26]. In order to ensure the compliance of the participants in the record of fluid intake, the investigators checked the questionnaire every day, and if there were mistakes in the records, the investigators would ask the participants to refill the questionnaire.

To assess the intake of water from food, all foods were weighted before and after the participants ate, which has been described in our previous study [26]. The duplicate portion method was used to determine the intake of water from food. All the samples of foods were measured according to the national standard of GB 5009.3-2016 [35], and the water from fruits or snacks was evaluated according to the China Food Composition Table (2009) [36]. The water from food was classified as follows: staple food (steamed bread, rice, etc.), dishes (vegetables, meat, fish and eggs), porridge (millet porridge and other porridges), soup (tomato egg soup and other soups) and snacks (fruits, nuts, etc.) [26].

### 2.6. Temperature and Humidity of the Environment

The indoor and outdoor temperature and humidity were measured during the study days (WSB-1-H2, Exasace, Zhengzhou, China). The places that participants were allowed access to include the dormitories and classrooms, allowing the microclimate both outdoors and indoors to be recorded. Therefore, the temperature and humidity of the study places indoors and outdoors were recorded every day at three time points, namely, 10:00 a.m., 2:00 p.m. and 8:00 p.m., for 7 days. The temperatures of indoors and outdoors during the 7 days were 21.8 °C and 20.7 °C, respectively. The humidity was 39.9% and 35.9%, respectively (see Appendix A).

### 2.7. Anthropometric Measurements

The height and weight measurements were performed twice with standard methods by trained investigators, which were described in detail in our previous study (HDM-300; Huaju, Zhejiang, China; Accu Measure, Greenwood Village, CO, USA) [26]. (BMI: weight (kg)/height squared (m^2^)).

### 2.8. Urine Collection and Measurements

Participants were given the urine containers designed by the researchers to collect the 24 h urine samples, including the first morning urine. After collection, all urine samples were stored at +4 °C before being determined. The urine volume, osmolality, USG, pH, urea and creatinine, and the concentrations of the electrolyte including the Na, K, Cl, Ca, phosphate and Mg, were determined as described in our previous study [29]. Furthermore, three methods were used to improve the collection of urine. First, the participants recorded information including the unique index of the participant, the void and time of each urine event on the urine container. Second, they recorded the related information on the urine questionnaire. Third, the investigators who responded for the measurements also recorded the related information for each urine sample the participants sent to the lab. Optimal hydration was defined when urine osmolality ≤500 mOsm/kg; middle hydration was defined as 500 mOsm/kg < urine osmolality ≤ 800 mOsm/kg; and dehydration was defined as urine osmolality >800 mOsm/kg [37].

### 2.9. Plasma Collection and Measurements

Fasting blood samples were collected for the determination of the osmolality and electrolyte concentrations, such as the Na, K, Cl, Ca, phosphate and Mg, which were measured with standard procedure, as described before [29] (AU 5800; Beckman, Brea, CA, USA).

### 2.10. Calculation Formulas Used

To estimate the salt intake, the following formulas were used [11]:$$\mathrm{Salt}\;(g)=\left(2.54\times 23\times 24\;h\;\mathrm{urinary}\;\mathrm{Na}\;\left(\frac{\mathrm{mmol}}{L}\right)\times 24\;h\;\mathrm{volume}\left(L\right)\right)÷1000$$

### 2.11. Statistics

The statistical analysis was assessed by the SAS 9.2 software (SAS Institute Inc., Cary, NC, USA). The results were reported as mean (95% CI (confidence interval)) if the data were normally distributed. If not, then, the median and quartile ranges (M and Q) were used to reveal the data. Participants were split into four groups: low salt intake 1 (LS_1_), low salt intake 2 (LS_2_), high salt intake 1 (HS_1_) and high salt intake 2 (HS_2_) groups, according to the quartiles of salt intake of the participants (Q1, 3.7–9.4 g; Q2, 9.5–12.3 g; Q3, 12.4–17.4 g; Q4, 17.5–33.6 g). The salt intakes were 7.6, 10.9, 14.7 and 22.4 g for the LS_1_, LS_2_, HS_1_ and HS_2_ groups, respectively, and differed significantly when comparing the two groups (all *p* < 0.05). Differences in the normally distributed data (reported as mean ± SD), such as the age, height, weight and BMI were compared using one-way ANOVA among the four groups; while, the Kruskal–Wallis H-test was used to compare the differences in the abnormal distribution data (shown as M and Q) among the four groups; chi-square test was used to compare the proportions of participants who met the adequate fluid intake of China, who met the recommendation of TWI of China and with optimal hydration status among the four groups. Differences between the two groups were compared using Student–Newman–Keuls (SNK) (*p* < 0.05). Significance levels were set at 0.05 (*p* < 0.05).

## 3. Results

In total, 159 participants were recruited, and 156 participants including 80 males and 76 females completed the periods of the study, which was a 98% completion rate.

Table 1 shows that the mean height and weight differed significantly when comparing LS_1_ and LS_2_ with HS_1_ and HS_2_, respectively (*p* < 0.05), with no differences between LS_1_ and LS_2_, or between HS_1_ and HS_2_. Furthermore, there were no significant differences in age, BMI, systolic pressure or diastolic pressure (all *p* > 0.05).

### 3.1. Measurement of Water Intake Patterns

Table 2 shows that the TWI and water from food all increased with the increase of salt intake (all *p* < 0.05). Participants in the HS_2_ group drank 310–381, 250–358 and 382–655 mL more water from total drinking fluids, water from food and TWI, respectively, than those in the groups of LS_1_ and LS_2_ (*p* < 0.05). However, no statistically significant differences in the contributions of total drinking fluids to TWI or water from food to TWI were found among the four groups (*p* > 0.05). Furthermore, there were no significant differences in the percentages of participants who met the recommendation of adequate total drinking fluids and TWI of China among the four groups (all *p* > 0.05).

### 3.2. Drinking Patterns

Table 2 shows that in the four groups, water was the main contributor of total drinking fluids, which accounted for 77.8–86.1%, and there were no significant differences among the four groups (*p* > 0.05). The amounts of water differed significantly in the four groups (*p* < 0.05), with 252–403 mL more water in HS_2_ and HS_1_ than in LS_1_ and LS_2_. Moreover, the intakes of SSBs in the HS_2_ group were lower than those in the LS_2_ and HS_1_ groups. Nevertheless, among the four groups, no statistically significant differences in the amounts of milk and milk products, tea, alcohol and even other beverages were found (*p* > 0.05). Otherwise, SSBs were the second contributor to total drinking fluids, followed by water in the groups of LS_1_ and HS_2_, but in the groups of LS_2_ and HS_1_, milk and milk products followed water. 

### 3.3. Water from Food

The current study showed that participants with different salt intake had different intakes of food, which increased from 1292 g in LS_1_ group to 1859 g in HS_2_ group (F = 24.816, *p* < 0.001). Participants in the HS_2_ group had 250–358 mL more amounts of water from food than those in groups of LS_1_ and LS_2_. The water from food was mainly from dishes and staple foods in all four groups. Participants in the HS_2_ group had 114, 228 and 171 mL more water from staple food, dishes and soup, respectively, than those in LS_1_ group, and 70 mL less water from porridge (*p* < 0.05). As for the sources of water from different types of food, the amounts of water from snacks did not differ significantly among the four groups (*p* > 0.05). About the contributions of the water from different types of foods, the percentages of water from soup and porridge in the water from food were different among the four groups (*p* < 0.05), as shown in Table 2.

### 3.4. Measurement of Urinary and Plasma Indexes

Table 3 reveals that the volumes of urine increased simultaneously from the LS_1_ group to the HS_2_ group, and the concentrations of Na, K, Cl and pH of urine were significantly different among the four groups (all *p* < 0.05). The percentage of participants with optimal hydration status was statistically higher in the LS_1_, LS_2_ and HS_1_ groups than in the HS_2_ group (*p* < 0.05). Regarding the plasma biomarkers, the concentrations of Cl and phosphate differed significantly among the four groups (*p* < 0.05). No statistically significant differences were found in other plasma biomarkers (*p* > 0.05), as shown in Table 4.

## 4. Discussion

In the present study, the average consumption of salt intake among the young adults was 13.9 g, which was greater than the recommendation of 6 g from the Chinese Nutrition Society, and also higher than the results of a survey conducted previously in China [38]. Furthermore, the salt intake among the four groups was significantly different between each other, amounting to 7.6, 10.9, 14.7 and 22.4 g in the LS_1_, LS_2_, HS_1_ and HS_2_ groups, respectively. In our study, when the salt intake increased significantly, the intakes of total drinking fluids, water from food and TWI increased simultaneously among the four groups. Meanwhile, to our surprise, the proportions of both the total drinking fluids and water from food in TWI did not change with the increase in salt intake. This finding is in line with a study examining the association between dietary salt and water intake. It has been demonstrated that fluid intake was transiently increased with increasing salt intake [13]. The findings revealed that young adults with higher salt intake may partake in a greater consumption of water from food; therefore, the amounts of the TWI were higher than their counterparts. The results were consistent with those of a cross-sectional study in which the dietary sodium intake was correlated with the fluids intake among US children and adolescents aged 2–18 years [18]. Some experimental studies in animals also agree with the results of our study [15,16]. However, a randomized repeated-measures experiment conducted among healthy young men showed that the acute intake of sodium did not affect thirst or water intake [39]. Another study in adult rats showed that 10 days of dietary sodium deficiency inversely increased water intake [40]. Therefore, more studies should be implemented to explore the internal mechanism of the regulation between water and salt in the human body. However, sometimes, the effects of the intake of fluids that include salt may be better than taking in water only. The results of the study demonstrated that healthy men had greater sodium balances after drinking beverages other than water [41]. The regulations of water and sodium should be clarified.

Regarding the amounts and contributions of fluid intake, water was shown to be the main contributor to total drinking fluids among the four groups, with proportions ranging from 77.8% to 88.0%, which were consistent with our previous study [29]. Regarding the types of beverages, SSBs, followed the water, were the second contributor in total drinking fluids among participants in groups of LS_1_ and HS_2_, contributing to nearly 10% of total drinking fluids; nevertheless, milk and milk products were the second contributor among participants in LS_2_ and HS_1_ groups, even though the amounts of milk (51–54 mL) were far less than the recommendation (300 mL) of the Chinese Nutrition Society. This observation suggests that participants with lower salt intake may not have healthier drinking behaviors than those with relative higher intake of salt. In the current study, water from food was an important source of TWI, accounting for 50.7–51.5% among the four groups. Concerning the sources of water from different types of food, the dishes (including the vegetables) and the staple foods (including the steamed bread) were the first and second contributors among the four groups; the amount of water from dishes and staple foods increased gradually with salt intake. Participants in the HS_2_ group had higher amount and proportion from soup, but lower amount and proportion of water from porridge than those in the LS_1_ group. From these observations, it seems that subjects with higher salt intake may tend to get more water from food.

The present study demonstrated that the 24 h urine volume and the concentrations of electrolyte including Na, K and Cl in urine increased with salt intake. Similarly, a recent systematic review including 21 randomized control trials which enrolled a total of 293 healthy adults revealed that higher dietary sodium increased the urine volume [42]. A recent study including one randomized controlled trial and one cross-sectional trial conducted to determine the association between the salt intake and volume of urine in adults showed that the 100 mmol/d reduction in the intake of salt decreased 24 h urine volume, with a reduction of 367 mL [43]. Evidence from one animal study conducted in lactating dairy cattle was concordant with the studies, reporting a positive linear relationship between sodium intake and urine volume [44]. However, a study including a total of 1339 Swiss participants revealed that the urine volume of the participants with different salt intakes was exactly the same [45]. Similarly, the DASH-Na study did not see any significant difference in the urine volume between adults with different salt intakes [46]. The methods of intervention and the collection of the urine influenced the differences mentioned above, and in the future, more studies are needed in this regard. Furthermore, the results of this study revealed that significantly more participants with 24 h urine osmolality lower than the standard were in the LS_1_, LS_2_ and HS_1_ groups compared to the HS_2_ group, which indicated that participants with higher salt intake were perhaps dehydrated, whereas participants with lower salt intake may have optimal hydration status. These findings suggest that high salt intake in free-living conditions among young adults should be avoided to prevent dehydration and the related adverse effects on health. However, the results of the DONALD study were not consistent with our study, reporting that the hydration status of healthy children and adolescents with a Western lifestyle was not influenced by salt intake [18]. The differences in the measurement of hydration and the ages of the participants could explain the observation.

Sodium was the principle constituent of plasma osmolality, which was also dependent on water intake and on water and sodium excretion through urine and sweat. In human beings, the osmolality of plasma is principally maintained at 280–295 mOsm/kg, through the integration of thirst, vasopressin secretion and renal responsiveness to vasopressin. In the present study, the plasma osmolality of young adults was 298–300 mOsm/kg, and the concentrations of Na did not differ significantly among participants with different levels of salt intake, which indicated that osmolality and the concentration of Na in plasma were not sensitive to the difference in salt intake in free-living conditions, similar to a previous study. Another study demonstrated that the acute effects of salt intake on the blood pressure were mediated by the osmolality [47]. In addition, another study exploring the acute effects of sodium content in beverages on the decrease in plasma sodium among healthy females observed that the concentration of sodium in plasma increased after sodium content beverage supplementation [48]. Furthermore, among the chronically salt-loaded mice and humans, the ability of the kidney and the generation of free water were more than sufficient to maintain the stability of the plasma osmolality [12,49].

A randomized clinical trial to explore the effect of a high-salt diet on endothelial microvascular function among healthy young adults compared to a low-salt diet showed that the 7-day high-salt diet impaired microvascular reactivity by affecting its endothelium-dependent vasodilation [50]. There is obvious evidence that the decrease in salt intake from the level of approximately 9–12 g/d at present in most countries to the recommended intake of 5–6 g/d lowered the blood pressure (BP) in adults with both hypertension and normal blood pressure [51]. In the present study, even though the blood pressure of the young adults was not higher than the standard, adverse effects of an excessive salt diet on health would appear if the participants did not change their dietary habits. Therefore, nutrition interventions such as decreasing the intake of salt in the free-living conditions should be considered.

Regarding the strengths of this study, to our knowledge, it was the first one to investigate the drinking patterns and hydration biomarkers among young adults with different salt intakes in China. Moreover, 24 h urine was collected each time participants urinated, using specific containers for 3 consecutive days, to diminish fluctuating factors during weekdays and weekends. Finally, the “gold standard” for determining the salt intake—the 24 h urine collection method [52]—was used to monitor the salt intake, which is the recommended method of the WHO [53]. Despite the advantages mentioned above, the study did have some limitations. The effects of the different levels of salt intake on urinary aquaporin-2 excretion were not investigated in this study. Secondly, the energy intake of the participants was not measured in the current study.

## 5. Conclusions

Participants with higher salt intake had higher amounts of total drinking fluids, water from food and TWI than those with lower salt intake. They also had inferior hydration status than those with lower salt intake. Interventions should be taken to reduce the consumption of salt among young adults.

## Figures and Tables

**Figure 1 nutrients-14-00287-f001:**
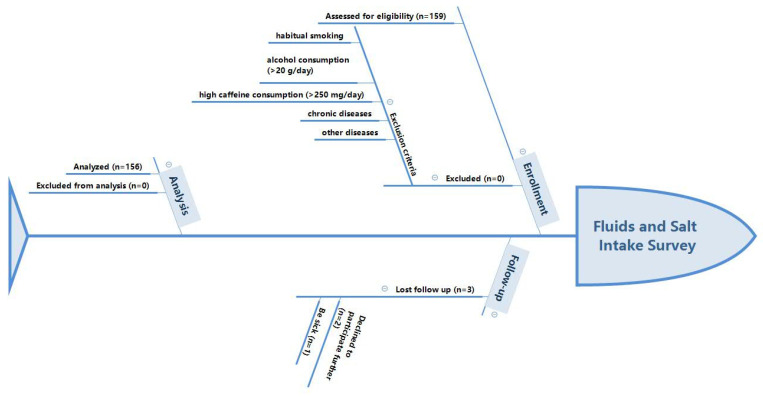
Participants flow chart.

**Figure 2 nutrients-14-00287-f002:**
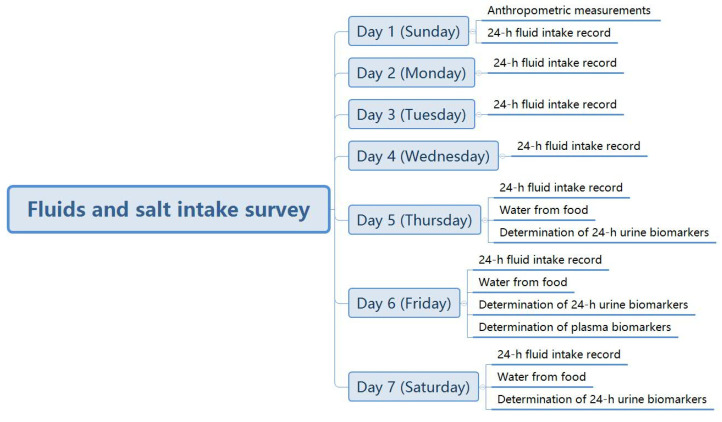
The study procedure.

**Table 1 nutrients-14-00287-t001:** The characteristics of participants.

	LS_1_ (*n* = 39)	LS_2_ (*n* = 39)	HS_1_ (*n* = 39)	HS_2_ (*n* = 39)	Total (*n* = 156)	*p*
Age (y)	19.7 ± 1.0	20.0 ± 1.1	19.8 ± 1.0	19.7 ± 1.2	19.8 ± 1.1	0.792
Height (cm)	163.6 ± 8.3 ^a^	163.2 ± 8.0 ^a^	167.4 ± 8.2 ^b^	170.7 ± 6.6 ^b^	166.2 ± 8.3	<0.001
Weight (kg)	59.2 ± 11.4 ^a^	57.2 ± 8.2 ^a^	61.8 ± 8.5 ^a^	67.1 ± 14.7 ^b^	61.3 ± 11.5	0.001
BMI (Kg/m^2^)	22.0 ± 3.1	21.4 ± 2.3	22.1 ± 2.6	23.0 ± 4.4	22.1 ± 3.3	0.219
Systolic pressure	115 ± 14	111 ± 15	114 ± 15	118 ± 20	115 ± 16	0.368
Diastolic pressure	69 ± 7	74 ± 10	72 ± 7	73 ± 9	72 ± 9	0.083

Note: Values are shown as the mean ± standard deviation (SD). BMI: body mass index. (a, b): The same symbol indicates that there were no statistically significant differences between groups; different symbols indicate that the differences were statistically significant between groups. In this table, the heights of participants in LS_1_ and LS_2_ were different from those of participants in other groups, and there were no differences between HS_1_ and HS_2_ groups. The weight of participants in LS_1_, LS_2_ and HS_1_ were different from those of participants in HS_2_ group, and there were no differences between LS_1_, LS_2_ and HS_1_ groups.

**Table 2 nutrients-14-00287-t002:** The TWI, total drinking fluids and water from food among participants.

	LS1 (*n* = 39)			LS2 (*n* = 39)			HS1 (*n* = 39)			HS2 (*n* = 39)			Total (*n* = 156)		
	M	Q	%	M	Q	%	M	Q	%	M	Q	%	M	Q	%
Total drinking fluids	962 ^a^	411	49.3%	1033 ^ab^	723	49.3%	1186 ^bc^	709	48.5%	1343 ^c^	648	49.2%	1135	620	50.6%
Meets the adequate fluid intake of China	36 (92.3%)			31 (79.5%)			30 (76.9%)			30 (76.9%)			127 (81.4%)		
Meets the recommendation of TWI of China	35 (89.7%)			32 (82.1%)			30 (76.9%)			28 (71.8%)			125 (80.1%)		
Water	726 ^a^	497	77.8%	745 ^ab^	684	84.0%	978 ^bc^	664	88.0%	1129 ^c^	646	86.1%	866	642	81.0%
Tea	0	0	0.0%	0	0	0.0%	0	0	0.0%	0	0	0.0%	0	0	1.0%
Milk and milk products	43	107	5.0%	54	162	5.6%	51	131	4.1%	36	114	3.0%	43	131	6.6%
SSBs	75 ^a^	176	8.1% ^a^	24 ^b^	64	2.5% ^ab^	0 ^b^	63	0.0% ^ab^	52 ^a^	188	4.3% ^ac^	43	112	8.0%
Alcohol	0	0	0.0%	0	0	0.0%	0	0	0.0%	0	0	0.0%	0	0	0.7%
Others	4	39	0.3%	0	45	0.0%	0	36	0.0%	9	30	0.7%	0	17	2.7%
Water from food	962 ^a^	284	50.7%	1070 ^ab^	318	50.7%	1281 ^c^	336	51.5%	1320 ^c^	318	50.8%	1174	373	49.4%
Staple food	252 ^a^	86	27.5%	285 ^ab^	161	28.2%	321 ^c^	151	25.9%	366 ^c^	125	28.2%	301	141	26.3%
Dishes	474 ^a^	185	50.8%	573 ^b^	190	52.6%	668 ^c^	189	53.5%	702 ^c^	267	54.5%	620	217	52.2%
Soup	15 ^a^	194	2.1% ^a^	75 ^a^	157	6.0% ^ab^	93 ^a^	186	7.5% ^ab^	186 ^b^	194	13.5% ^bc^	93	195	10.1%
Porridge	127 ^a^	176	13.3% ^a^	115 ^a^	123	10.3% ^a^	111 ^a^	259	11.3% ^a^	57 ^b^	100	4.1% ^b^	97	182	10.4%
Snacks	0	0	0.0%	0	0	0.0%	0	0	0.0%	0	0	0.0%	0	0	0.9%
Total water intake	1948 ^a^	648	_	2221 ^ab^	780	_	2529 ^bc^	740	_	2603 ^c^	376	_			_

Note: Values are shown as the median (M) and quartile ranges (Q); (a, b, c): The same symbol indicates that there were no statistically significant differences between the two groups; different symbols indicate that the differences were statistically significant between the two groups. For example, the superscripts in the column of amounts of total drinking fluids of participants in LS_1_, LS_2_, HS_1_ and HS_2_ groups are a, b, c and d, which means that the total drinking fluids are different between the groups; the superscripts in the column of volume of water from food of participants in the four groups are a, ab, c and c, which means that there are no differences between LS_1_ and LS_2_, HS_1_ and HS_2_, but significant differences between LS_1_ and HS_1_, and LS_2_ and HS_2_ were found. %: Contributions of total drinking fluids and water from food to TWI; percentages of different fluids in total drinking fluids; proportions of water from different foods in water from food. There were statistically significant differences in the amounts of TWI, total drinking fluids, water from food, plain water and SSBs (χ^2^ = 30.547, *p* < 0.001; χ^2^ = 13.670, *p* = 0.003; χ^2^ = 44.685, *p* < 0.001; χ^2^ = 15.268, *p* = 0.002; χ^2^ = 11.892, *p* = 0.008) among the four groups, respectively. There was statistical significance in the amounts of staple food, porridge, soup and dishes (χ^2^ = 29.953, *p* < 0.001; χ^2^ = 11.666, *p* = 0.009; χ^2^ = 16.896, *p* = 0.001; χ^2^ = 47.660, *p* < 0.001) among the four groups, respectively. There were no statistically significant differences in the contributions of total drinking fluids and water from food to TWI (*F* = 0.058, *p* = 0.982; *F* = 0.058, *p* = 0.982). There was statistical significance in the contributions of SSBs to total drinking fluids among the four groups (χ^2^ = 9.813, *p* = 0.020). The contributions of porridge and soup to water from food among the four groups were different (χ^2^ = 16.054, *p* = 0.001; χ^2^ = 10.330, *p* = 0.016).

**Table 3 nutrients-14-00287-t003:** The characteristics of 24 h urine among participants.

	LS1 (*n* = 39)				LS2 (*n* = 39)				HS1 (*n* = 39)				HS2 (*n* = 39)				*p*
	M	Q	X	SD	M	Q	X	SD	M	Q	X	SD	M	Q	X	SD	
24 h volume (mL)			1049 ^a^	479			1317 ^b^	468			1467 ^bc^	447			1522 ^c^	460	**<0.001**
24 h urine osmolality (mOsm/kg)	625	390			582	315			541	316			636	268			0.463
(≤500 mOsm/kg, n, %)	16 (41.0%) ^a^				16 (41.0%) ^a^				16 (41.0%) ^a^				10 (25.6%) ^b^				**0.032**
Na (mmol/L)	133 ^a^	62			151 ^ab^	78			192 ^b^	91			245 ^c^	105			**<0.001**
K (mmol/L)			33.3 ^a^	12.6			33.4 ^a^	14.8			32.6 ^a^	10.6			42.6 ^b^	14.2	**0.002**
Cl (mmol/L)	132 ^a^	64			143 ^ab^	79			189 ^b^	99			244 ^c^	95			**<0.001**
Mg (mmol/L)			2.83	1.21			2.64	1.16			2.40	0.92			2.66	0.89	0.340
Ca (mmol/L)	2.04	1.57			1.98	1.40			2.05	1.26			2.55	1.86			0.081
Phosphate (mmol/L)	14.93	13.21			15.22	10.83			14.21	6.89			16.18	10.45			0.254
Creatinine (mmol/L)			106.8	51.8			91.2	38.4			87.7	33.7			100.5	35.7	0.274
Uric acid (mmol/L)			26.4	9.5			24.3	11.1			23.1	9.2			26.4	9.1	0.354
Urea (mmol/L)			225.3	94.0			211.5	92.3			192.4	67.9			232.1	80.0	0.173
USG			6.6	0.3			6.7	0.3			6.8	0.2			6.7	0.3	0.295
pH			1.016 ^a^	0.004			1.016 ^ab^	0.004			1.015 ^ab^	0.004			1.017 ^b^	0.004	**0.011**

Note: Values are shown as the median (M) and quartile ranges (Q) or mean ± standard deviation (SD). (a, b, c): The same symbol indicates that there were no statistically significant differences between the two groups; different symbols indicate that the differences were statistically significant between the two groups. For the urinary biomarkers, there were statistically significant differences in the volume of urine, the hydration status and the concentrations of Na, K, Cl and pH of urine among the four groups (*F* = 8.152, *p* < 0.001; χ^2^ = 13.785, *p* = 0.032; χ^2^ = 56.472, *p* < 0.001; *F* = 5.088, *p* = 0.002; χ^2^ = 56.110, *p* < 0.001; *F* = 3.873, *p* = 0.011).

**Table 4 nutrients-14-00287-t004:** The characteristics of plasma biomarkers among participants.

	LS_1_ (*n* = 39)	LS_2_ (*n* = 39)	HS_1_ (*n* = 39)	HS_2_ (*n* = 39)	
	X (95% CI)	X (95% CI)	X (95% CI)	X (95% CI)	*p*
Osmolality (mOsm/kg)	299 [297, 301]	299 [297, 301]	299 [297, 301]	299 [298, 301]	0.935
Na (mmol/L)	141 [141, 141]	141 [141, 142]	141 [140, 141]	141 [140, 141]	0.241
K (mmol/L)	4.6 [4.5, 4.7]	4.5 [4.4, 4.7]	4.5 [4.4, 4.6]	4.5 [4.4, 4.7]	0.953
Cl (mmol/L)	103 [102, 103] ^a^	104 [103, 104] ^ab^	104 [104, 105] ^b^	104 [104, 105] ^b^	**0.025**
Ca (mmol/L)	2.52 [2.49, 2.52]	2.53 [2.50, 2.54]	2.51 [2.48, 2.54]	2.51 [2.49, 2.53]	0.781
Phosphate (mmol/L)	1.35 [1.31, 1.39] ^a^	1.33 [1.27, 1.38] ^a^	1.30 [1.25, 1.36] ^a^	1.22 [1.18, 1.27] ^b^	**0.002**
Mg (mmol/L)	0.92 [0.90, 0.94]	0.91 [0.89, 0.93]	0.92 [0.90, 0.94]	0.92 [0.90, 0.94]	0.770

Note: Values are shown as the mean (95% CI). (a, b): The same symbol indicates that there were no statistically significant differences between the two groups; different symbols indicate that the differences were statistically significant between the two groups. There were statistically significant differences in the concentrations of Cl and phosphate among the four groups (*F* = 3.189, *p* = 0.025; *F* = 5.181, *p* = 0.002).

## Data Availability

The data of this study is available from the corresponding author on reasonable request.

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
