# Peer review of "Young Adults with Higher Salt Intake Have Inferior Hydration Status: A Cross-Sectional Study"

_nutrients, 2022, doi:10.3390/nu14020287_

Round 1

Reviewer 1 Report

General:

Please work on a more precise language, including removing causal language based on the findings of this trial

Remove duplication sections

The manuscript needs a more detailed description of the regulation of water and salt in the human body, which will also benefit the discussion section

Causality cannot be inferred with this study design. Salt and water intake may not be independent of each other, which is often an assumption in many statistical tests. Check if sex and/or energy intake can explain some of your findings.

Introduction:

Define dehydration and overhydration in relation to the symptoms they may cause.

Definitions of key outcome variables are needed. 

The objective are not clearly operationalized. if you want to investigate more than one problem, you should describe each.

Methods:

How were the exclusion criteria operationalized?

Sample size calculations are lacking. How did you end up with 160 participants?

How does the fluid intake questionnaire look like? Validity measures?

The optimal hydration was defined when urine osmolality ≤ 500 mOsm/kg;  

Reference for calculation of salt intake in g is needed

Describe variables collected/calculated

None of the groups have a low salt intake. New names?  How did you use the repeated samples?

Elaborate on statistical methods. What methods for mean/ for median/ other distributions.

How was the salt intake calculated across 7 days?

Results:

Where all background variables normally distributed – table 1?

Line 226 ref is missing?

Isn't the osmolality generally very high in your sample? how were your lab-quality measures?

Average… use mean or median

Is it possible to describe CI to better indicate the uncertainty of estimates?

Discussion

Could the observations and associations simply be due to different SES status and/or gender that affect food and drink intake? (PORRIDGE DIFFERENCES inverse of salt intake. Does porridge correlate with SES?). ;

You say: "The amount of water from dishes and staple foods increased gradually with salt intake". OR could it be: As people eat more, they also get more salt through the food…

Are animal references relevant for these findings? new references?

In my opinion, the most important finding are the generally high salt intake - which is of public health concern. It is fine to describe associations, but you may discover that you just discover "the two sides of the same coin"

I just quickly highlighted sentences or words that you need to look more closely at, having my general suggestions in mind

Author Response

Comments and Suggestions for Authors

General:

Please work on a more precise language, including removing causal language based on the findings of this trial

Response: Thanks for your comments. We have revised accordingly in the manuscript.

The manuscript has been carefully revised by MDPI (English edited 36551), in order to make sure that there were no errors in the English language and grammars throughout the manuscript.

Remove duplication sections

Response: Thank you.

We have made the revision according to your comments.

We removed the duplication sections of the manuscript according to the duplication report.

The manuscript needs a more detailed description of the regulation of water and salt in the human body, which will also benefit the discussion section

Response: Thanks for your comments. It has been revised accordingly (Lines 50-56, 74-76, Page 2).

Causality cannot be inferred with this study design. Salt and water intake may not be independent of each other, which is often an assumption in many statistical tests. Check if sex and/or energy intake can explain some of your findings.

Response: Thank for your comments. We had added into the Introduction Section (Lines 50-56, 74-76, Page 2).

In one study investigating the associations between water intake and salt intake among children and adolescents aged 2-18 year, showed that dietary salt intakes were all greater in males than females in all the age groups. In the present study, males and females were divided into four groups, which means that in each group, there would be males and females and the effects of sex on the salt intake may not explain the differences of the water intake patterns and hydration biomarkers among the four groups. In the future, more related model that used to investigate the associations between the salt intake and water intake patterns should be determined.

As for the energy intake, it was not collected in the present study, therefore, we did not explore the effects of the energy intake on the water intake patterns and hydration biomarkers. In the future, the energy intake should be collected.

Introduction:

Define dehydration and overhydration in relation to the symptoms they may cause.

 Response: Thanks for your comments. It has been revised accordingly (Lines 43-46, Page 1).

Definitions of key outcome variables are needed. 

 Response: Thanks for your comments. It has been revised accordingly (Lines 95-101, Pages 2-3).

The key outcomes of our study were the water intake patterns (including the drinking pattern and the food intake pattern) and the hydration biomarkers. All the definitions of them had been described in detail in the manuscript (Lines 95-101, Page 2).

In the present study, we wanted to explore the differences in the water intake patterns, which including the drinking patterns (defined here as the types and amounts of fluids intake here), the food intake pattern (defined here as the amounts of water from different types of food consumed) and the hydration biomarkers including the urinary and plasma biomarkers among participants with different levels of salt intake. 

The objective are not clearly operationalized. if you want to investigate more than one problem, you should describe each.

 Response: Thanks for your comments. We have revised accordingly in the Introduction Section, dividing the aims into two parts, with the first one was to investigate the differences in the drinking patterns and the food intake patterns in young adults with different levels of salt intake in free-living conditions, and the second one was to examine the differences of and hydration biomarkers among them. (Lines 117-121, Page 3).

Methods:

How were the exclusion criteria operationalized?

Response: Thanks for your comments.

Before participating in our research, the health situations of the subjects were measured using the self-designed health-related questionnaire. The information such as the habit of smoking, alcohol intake, the gastrointestinal diseases, family history of diseases were collected by the questionnaire.

Furthermore, they were also asked to have ECG (electrocardiogram), urine test and plasma test. The urinary tests including the urine routine, the urine biochemistry were observed. The levels of glucose, urea and creatinine of blood were tested using the fasting plasma.

The questionnaire and the tests were performed to make sure that the participants were healthy.

Sample size calculations are lacking. How did you end up with 160 participants?

Response: Thank you. The sample size calculation was added into the Methods Section, that “According to a study conducted among adults [28], the detection rate (p) of the adults with the intake of salt exceeded the recommendation of Chinese dietary guide-lines of 6 g/d, was 0.7, then, set the d=0.11, α=0.05, Zα=1.96, the sample size was calculated using the formula as followed: n= Z2αp (1-p)/d2

Totally, 136 of participants were in need. In the present study, 159 participants were recruited.” (Lines 126-133, Page 3).

How does the fluid intake questionnaire look like? Validity measures?

Response: Thanks for your comments.

The fluids intake questionnaire was self-designed, which was used to collect the information of the time, place, type and the amounts of the fluid intake every time of the participants.

The questionnaire had been used in our previous studies, which was used in the two large surveys. One study was conducted among 1483 adults from four cities of China in summer and another study was performed among 5848 children and adolescents in four cities of China. The validity and accuracy of the questionnaire were high.

Furthermore, in the present study, in order to ensure the compliance of the participants in the record of fluid intake, the investigators checked the questionnaire every day, and if there were mistakes in the records, the investigators would ask the participants to refill the questionnaire.

The studies that used the questionnaire were showed as follow:

[1] Ma G, Zhang Q, Liu A, et al (2012) Fluid intake of adults in four Chinese cities. Nutr Rev 70:S105–S110.

[2] Du S, Hu X, Zhang Q, et al (2013) Water intake of primary and middle school students in four cities of China. Chin J Prev Med 3:210–213.

[3] Zhang N, Du S, Tang Z, et al (2017) Hydration, fluid intake, and related urine biomarkers among male college students in Cangzhou, China: a cross-sectional study-applications for assessing fluid intake and adequate water intake. Int J Environ Res Public Health 14:513.

[4] Zhang J, Zhang N, Liang S, et al (2019) The amounts and contributions of total drinking fluids and water from food to total water intake of young adults in Baoding, China. Eur J Nutr 58:2669-2677.

The optimal hydration was defined when urine osmolality ≤ 500 mOsm/kg;  

Response: Thanks for your comments.

The hydration status was determined by the 24h urine osmolality: The optimal hydration was defined when urine osmolality ≤ 500 mOsm/kg; middle hydration was defined as 500 mOsm/kg < urine osmolality ≤ 800 mOsm/kg; and dehy-dration was defined as urine osmolality > 800 mOsm/kg.

References:

[1] Zhang N, Du S, Tang Z, et al. Hydration, fluid intake, and related urine biomarkers among male college students in Cangzhou, China: a cross-sectional study—applications for assessing fluid intake and adequate water intake. Int J Environ Res Public Health. 2017;14(5): 513.

[2] Perrier E, Vergne S, Klein A, et al. Hydration biomarkers in free-living adults with different levels of habitual fluid consumption. Br J Nutr. 2013;109(9): 1678–87.

Reference for calculation of salt intake in g is needed

Response: Thanks for your comments. The reference of the calculation of salt intake was added into the Methods Section (Line 203, Page 6).

Describe variables collected/calculated

Response: Thank you. We have made revision according to your comments (Lines 153-203, Pages 4-6).

None of the groups have a low salt intake. New names?  How did you use the repeated samples?

Response: Thanks for your comments. We have revised the name of the groups into Low salt and High salt intake groups, which were LS1, LS2, HS1 and HS2, respectively (Lines 208-209, Page 6).

Before the beginning of the project, we wanted to investigate the water intake patterns and hydration biomarkers among participants, furthermore, we also wanted to investigate the relationship between the salt intake and water intake among the young adults. After pre-investigation among the young adults, and the results of the study conducted among young males, we found that the salt intake of them were high than the recommendation of Chinese dietary guidelines. We wonder if the salt intake influenced the water intake patterns and the hydration biomarkers of them. After calculating the sample size, 159 and 136 were the participants that needed for the aims mentioned above. Therefore, we recruited 159 participants for the researches.

Elaborate on statistical methods. What methods for mean/ for median/ other distributions.

Response: Thanks for your comments. We have made the revision accordingly in the Methods Section.

It has been showed as followed “Differences in the normally distributed data (reported as mean ± SD), such as the age, height, weight and BMI were compared using One-way ANOVA among the four groups; Kruskal–Wallis H-test was used to comparing the differences in the abnormal distribution data (shown as M and Q) among the four groups; Chi-square test was used to comparing the proportions of participants meet the adequate fluid intake of China, meet the recommendation of TWI of China and with optimal hydration status among the four groups” (Lines 206-219, Page 6).

How was the salt intake calculated across 7 days?

 Response: Thanks for your comments.

In the present study, we wanted to explore the differences in the water intake patterns, which including the drinking patterns (defined here as the types and amounts of fluids intake here), the food intake pattern (defined here as the amounts of water from different types of food consumed) and the hydration biomarkers including the urinary and plasma biomarkers among participants with different levels of salt intake. Therefore, we should know the information about the salt intake and the fluids intake of the participants.

As known, the 24-h urinary sodium excretion is the gold standard for estimating sodium intake. In our study, three days (two weekdays and one weekend) of 24h urine were collected. The concentrations of electrolytes including the Na were measured.

Moreover, the 7 day 24h fluids intake record questionnaire was the gold standard for assessing the fluids intake of participants. Then, the fluids intake was collected for 7 days.

Results:

Where all background variables normally distributed – table 1?

Response: Thanks for your comments.

All the anthropometric measurements were normally distributed in our study, which could be explained by the following that all the participants were recruited from the one college in Hebei, China, and all of them were freshmen or sophomores, with the similar age.

Line 226 ref is missing?

Response: Thanks for your comments. We have added the reference into the manuscript (Line 202, Page 5).

Isn't the osmolality generally very high in your sample? how were your lab-quality measures?

Response: Thanks for your comments.

Urine samples were stored at 4 â—¦C in a refrigerator prior to assessments.

In order to make sure that all the urines were collected, we took four measures to improve quality control. The first one was that, participants were asked to record the information of every urine, including the time and the voids, on the questionnaire of “3-day-24-h urinary behavior questionnaire”. They also should write the same information on the urine container. Moreover, they should give the questionnaire to the investigators every day. The second one was that, when the samples of urine were sent to the laboratory, the researchers were asked to record the information of the urine, including the time and the voids of every participant, into the questionnaire that designed for them. The third one was that, researchers should compare the two records to find if there were some differences. If there were differences, researchers should check the questionnaire and the information on the container, to correct the errors. The fourth one was that, we also compared the information of the urine and the information of total drinking fluids to make sure that all the samples of urine were recorded.

Urine osmolality was assessed with freezing point method by experienced investigators with osmotic pressure molar concentration meter (SMC 30C; Tianhe, Tianjin, China). In order to make sure that all the osmolality was measured accurately, we took four measures to improve quality control. Firstly, before measuring the samples, the calibration of the instrument according to the procedure were performed. Secondly, each urine was coupled with 2-3 parallel samples and if the data of the urine was very high or low, and were not consistent with the color of the urine, the investigators should report to the researchers and measured the sample twice to find out the reason. Thirdly, the researchers also take out 5 samples for every 20 samples to test again and to compare the data measured before.  

The osmolality was high in the present study, which indicating that the most of the participants were not with optimal hydration status, which were consistent with the results of our previous study conducted among young males, in which the osmolality of the urine among was 653 mOsm/kg. Therefore, interventions should be proposed to increase the intake of fluids among the young adults.

Average… use mean or median

Response: Thanks for your comments. We have made the revision accordingly, that removed the “average”.

Is it possible to describe CI to better indicate the uncertainty of estimates?

 Response: Thanks for your comments. We have made the revision accordingly.

The data that normally distributed was showed as mean (95% CI).

Discussion

Could the observations and associations simply be due to different SES status and/or gender that affect food and drink intake? (PORRIDGE DIFFERENCES inverse of salt intake. Does porridge correlate with SES?). ;

Response: Thanks for your comments.

The study conducted among children and adolescents, demonstrated that the males had greater intake of salt than females, and subjects with different socioeconomic status had different intake of salt.

However, in the current study, firstly, each group had the males and females in each group, therefore, the gender may not affect the water intake patterns and hydration biomarkers. Moreover, the porridge was one of the common species of the breakfast for Chinese, which was composed by rice or millet and some vegetables in the present study. Therefore, it may not be related with the SES.

You say: "The amount of water from dishes and staple foods increased gradually with salt intake". OR could it be: As people eat more, they also get more salt through the food…

Response: Thanks for your comments.

The result of the study showed that the participants with higher salt intake had higher amounts of water from food than their counterparts. After analysis the amounts of the food that the participants took, it demonstrated that participants with higher salt intake had higher amounts of food, increasing from 1292 g in LS1 group to 1859 g in HS2 group (Lines, Page). Therefore, we could say that as people eat more, they also get more salt through the food. But the aim of the current study was to investigate the differences in the food intake patterns which including the types and amounts of the water from food. Then, we concluded that the amount of water from dishes and staple foods increased gradually with salt intake.

Are animal references relevant for these findings? new references?

Response: Thanks for your comments.

There were references relevant to the findings of our studies and we compared the results in the Discussion Sections.

We search the articles lately but with no new reference were found. And we will continue to pay attention.

In my opinion, the most important finding are the generally high salt intake - which is of public health concern. It is fine to describe associations, but you may discover that you just discover "the two sides of the same coin"

Response: Thanks for your comments.

The sale intake of the four groups were 7.6, 10.9, 14.7 and 22.4 g, respectively, which were much higher than the recommendation of sodium among Chinese. High sodium or salt intake has also been linked with high risk of cardiovascular diseases in the general population. Therefore, the salt reduction interventions should be proposed among the young adults in the future.

Bute in the current study, we aimed to know the differences of biomarkers among the four groups, the correlations between the water intake patterns, the hydration biomarkers and salt intake were not the aim of the study, so we did not perform the correlations. We analyzed the correlations in the other article in details.

I just quickly highlighted sentences or words that you need to look more closely at, having my general suggestions in mind

Response: Thanks for your comments. We have made the revision accordingly.

Reviewer 2 Report

Thanks for the opportunity to review.

I do have a concern with all the fluids being classified as water. It brings the validity of the study into question.

If published, here are comments:

Page 1 line 21 change h to hours, line 24 need to specify what LD is

Page 2 line 50 change high to increase risk, Line 53 capitalize The, Line 86 needs rewording 

Author Response

Reviewer 2

Open Review

English language and style

( ) Extensive editing of English language and style required
( ) Moderate English changes required
(x) English language and style are fine/minor spell check required
( ) I don't feel qualified to judge about the English language and style

Yes

Can be improved

Must be improved

Not applicable

Does the introduction provide sufficient background and include all relevant references?

( )

(x)

( )

( )

Is the research design appropriate?

( )

(x)

( )

( )

Are the methods adequately described?

( )

(x)

( )

( )

Are the results clearly presented?

( )

(x)

( )

( )

Are the conclusions supported by the results?

( )

(x)

( )

( )

Comments and Suggestions for Authors

Thanks for the opportunity to review.

I do have a concern with all the fluids being classified as water. It brings the validity of the study into question.

Response: Thanks for your comments.

All the drinking fluids were classified as water (plain water, tap water and bottled water), tea (fermented tea and semi-fermented tea), milk and milk products (liquid milk, yogurt and other milk products), SSBs (carbonated drinks, sports drinks, sweetened fruit juice, vegetable juice, protein drinks and other sugared drinks), alcohols (wine, beer, liquor, etc.) and other beverages. In order to ensure the compliance of the participants in the record of fluid intake, the investigators checked the questionnaire every day, and if there were mistakes in the records, the investigators would ask the participants to refill the questionnaire (Lines 158-160, Page 5).

If published, here are comments:

Page 1 line 21 change h to hours, line 24 need to specify what LD is

Response: Thanks for your comments. We have revised accordingly (Lines 22-24, Page 1).

Page 2 line 50 change high to increase risk, Line 53 capitalize The, Line 86 needs rewording 

Response: Thanks for your comments. We have made the revision accordingly (Lines 57, 62 and 97, Page 2).

Round 2

Reviewer 2 Report

Thanks for updating your submission. The quality of the paper is much improved.

Author Response

Authors' Response to Reviewers' Comments

Reviewer 2

Open Review

English language and style

( ) Extensive editing of English language and style required
( ) Moderate English changes required
(x) English language and style are fine/minor spell check required
( ) I don't feel qualified to judge about the English language and style

Yes

Can be improved

Must be improved

Not applicable

Does the introduction provide sufficient background and include all relevant references?

( )

(x)

( )

( )

Is the research design appropriate?

( )

(x)

( )

( )

Are the methods adequately described?

( )

(x)

( )

( )

Are the results clearly presented?

( )

(x)

( )

( )

Are the conclusions supported by the results?

( )

(x)

( )

( )

Comments and Suggestions for Authors

Thanks for updating your submission. The quality of the paper is much improved.

Response: Thanks for your comments.
